# Spectral Modulation of Optofluidic Coupled-Microdisk Lasers in Aqueous Media

**DOI:** 10.3390/nano9101439

**Published:** 2019-10-11

**Authors:** Zhihe Guo, Haotian Wang, Chenming Zhao, Lin Chen, Sheng Liu, Jinliang Hu, Yi Zhou, Xiang Wu

**Affiliations:** 1Key Laboratory of Micro and Nano Photonic Structures (Ministry of Education), Department of Optical Science and Engineering, Shanghai Engineering Research Center of Ultra Precision Optical Manufacturing, Fudan University, Shanghai 200433, China; 17110720004@fudan.edu.cn (Z.G.); 13110720009@fudan.edu.cn (C.Z.); 13110720001@fudan.edu.cn (L.C.); 14110720003@fudan.edu.cn (S.L.); 14110720001@fudan.edu.cn (J.H.); 18110720008@fudan.edu.cn (Y.Z.); 2Jiangsu Key Laboratory of Advanced Laser Materials and Devices, School of Physics and Electronic Engineering, Jiangsu Normal University, Xuzhou 221116, China

**Keywords:** microdisk laser, spectral modulation, optofluidic, coupled microdisk

## Abstract

We present the spectral modulation of an optofluidic microdisk device and investigate the mechanism and characteristics of the microdisk laser in aqueous media. The optofluidic microdisk device combines a solid-state dye-doped polymer microdisk with a microfluidic channel device, whose optical field can interact with the aqueous media. Interesting phenomena, such as mode splitting and single-mode lasing in the laser spectrum, can be observed in two coupled microdisks under the pump laser. We modulated the spectra by changing the gap of the two coupled microdisks, the refractive indices of the aqueous media, and the position of a pump light, namely, selective pumping schemes. This optofluidic microlaser provides a method to modulate the laser spectra precisely and flexibly, which will help to further understand spectral properties of coupled microcavity laser systems and develop potential applications in photobiology and photomedicine.

## 1. Introduction

Microcavity lasers based on whispering-gallery mode (WGM), possessing small size and strong optical confinement, have been attracting great interest in fundamental research and for practical applications [1,2,3,4]. Compared to the laser wavelength, the size of the microcavity is usually large enough to cause multimode lasing, resulting in a noisy spectrum, which severely degrades the performance of microlasers in applications such as optical communication and ultrasensitive optical sensors [5,6]. Coupled microcavity configuration is a compact and effective method to purify and regulate the laser spectra of microcavities. For example, a single-mode laser can be realized, based on the Vernier effect, in two coupled cavities with slightly different sizes [6,7,8,9,10,11]. Non-Hermitian optical systems, which have emerged recently, for lasers are, in essence, open and non-conservative systems. This means the most common gains and losses of physical quantities can be distributed differently with artificial purposes in such coupled microcavities [12]. In particular, if the optical microcavity is tuned to the exceptional point (EP) [13], where the resonance frequencies of the coupled modes coalesce both in their real and imaginary parts and become degenerate, some unique phenomena appear, such as parity–time (PT) symmetry breaking [14,15,16,17,18], chiral modes [19], and enhanced perturbation at higher-order EPs [20,21]. Therefore, this principle provides an alternative approach to modulate the laser spectrum by controlling the gain and loss elements. By managing the gain and loss distribution for the coupled modes in microcavities, some studies have successfully achieved single-mode lasing in WGM microcavities [22,23,24].

One of the key points in such coupled microcavity systems is the flexible method for controlling the system parameters, for example, the coupling strength between the coupled laser modes. The coupling strength can be regulated by controlling the gap between different microcavities, which is predetermined for the desired modes in the fabrication process, or precisely controlled by the nano-positioning stages in real-time experiments [14,25]. However, these mechanisms all lack efficiency and flexibility, and are sensitive to external noise sources, such as acoustic waves. Moreover, the coupled cavities fabricated on chip have the fixed coupling gaps. They need a reliable and flexible means for controlling the coupling strengths on chip. Optofluidic devices can integrate some functional fluids into optical or photonics devices [26,27]. Such fluids usually have high transmittance and tunable optical properties, e.g., the refractive index [28,29]. Because the optofluidic devices have a good compatibility with aqueous media, they are widely used in biology and medical science [30,31,32,33,34]. Therefore, optofluidic devices have found great potential in on-chip microlaser systems for their highly integrated level and, more importantly, the microlaser cavity is given a liquid environment, which provides an additional means for tuning the microlaser. 

In this study, solid-state dye-doped polymer microdisks are fabricated and integrated with a microfluidic channel, in order to investigate the mechanism and characteristics of the microdisk laser in aqueous media. Some phenomena, such as single-frequency laser and mode splitting in the coupled microcavities, were studied theoretically and experimentally in aqueous media for the first time. We observe that the resonance detuning of mode splitting is affected by changes in the gap between two coupled microdisks. Mode splitting and single-mode lasing in the laser spectrum can also be modulated by changing the refractive indices of the aqueous media and the position of the pump light. In addition, the aqueous media not only have a spectral modulation function but also a mode purification ability because they decrease the refractive index difference between the cavity and the background media, which eliminates some high-order WGM with larger reflection angles at the cavity surface. The results presented in this paper show that optofluidic microlasers possess potential for several promising applications, such as tunable single-mode lasers on chip and bio-photonic sensors.

## 2. Coupled-Mode Analysis

The coupled-mode theory could be applied to analyze such a coupled-microdisk laser system. We assume the two coupled modes in two respective cavities have amplitudes *a* and *b*. Considering the sinusoidally varying fields in the time domain, $e^{-i\omega_{n}t}$, *a* and *b* obey the coupled differential equation [20]:(1)$$i\frac{d}{dt}\left(\begin{array}{c} a\\ b \end{array}\right)=\left(\begin{array}{cc} ig_{a} & \kappa \\ \kappa & ig_{b} \end{array}\right)\left(\begin{array}{c} a\\ b \end{array}\right),$$
where *g_a_* and *g_b_* represent the gain (*g* > 0) or loss (*g* < 0) for modes *a* and *b*, respectively. *κ* denotes the coupling strength between the two microcavities. It leads to an eigenvalue problem: *H*_0_***V_n_***
*= ω_n_**V_n_***, where the eigenvalue *ω_n_* and eigenvector ***V_n_*** satisfy ***V_n_***
$e^{-i\omega_{n}t}$ = (*a*, *b*)^T^, and the Hamiltonian *H*_0_ of the system is
(2)$$H_{0}=\left(\begin{array}{cc} ig_{a} & \kappa \\ \kappa & ig_{b} \end{array}\right),$$

To determine the eigenfrequency, we solve the equation det(*H*_0_ – *ω_n_I*) = 0 (where *I* is the identity matrix), which results in two eigenfrequencies:(3)$$\omega_{\pm }=\frac{i(g_{a}+g_{b})}{2}\pm \sqrt{\kappa^{2}-\frac{{\left(g_{a}-g_{b}\right)}^{2}}{4}},$$

The difference between the two eigenfrequencies is
(4)$$\Delta \omega \equiv \omega_{+}-\omega_{-}=2\sqrt{\kappa^{2}-{\left(g_{a}-g_{b}\right)}^{2}/4},$$
where $\Delta \tilde{g}=\left(g_{a}-g_{b}\right)/2\kappa$ is the normalization gain difference between the two modes. If the two modes experience the same gain or loss, i.e., $\Delta \tilde{g}=0$, the frequency difference equals 2*κ*, which is the well-known mode-splitting effect, as shown in Figure 1a. However, the two modes have different gains or losses in more general situations. In particular, such frequency difference changes into a purely imaginary number when $\Delta \tilde{g}>1$. In this case, the difference between the two eigenfrequencies no longer shows two different resonance wavelengths in the spectrum, but instead, introduces two modes with different growth or decay rates, which often indicate different net gains in a laser system. Moreover, at the threshold where $\Delta \tilde{g}=1$ two eigenfrequencies coalesce to a single one, with *ω* = *i*(*g_a_* + *g_b_*)/2.

The coupling strength *κ* could be related to the coupling coefficient *K*; defined in the coupling between two parallel waveguides:(5)$$\kappa =\frac{KcL_{eff}}{n_{eff}2\pi R},$$
where *L_eff_* and *n_eff_* are the effective interaction length and effective index of the cavity modes, respectively, and *R* is the radius of the microdisk. The coupling coefficient *K*; can be expressed by the overlap integral between the two coupling modes:(6)$$K=-\frac{i\omega }{4}\int_{-\infty }^{+\infty }\int_{-\infty }^{+\infty }\left(n_{0}{}^{2}-n^{2}\right)E_{a}•E_{b}dxdy,$$
where ***E_a_*** and ***E_b_*** are the normalized field distributions in the two respective cavities, and *n*_0_ and *n* are the refractive indices for the microcavities and the background medium, respectively. The coupling strength *K* in Equation (6) can be calculated numerically, and the relationships between the coupling strength *K* and refractive index *n* under different gaps are shown in Figure 1b. *K* grows with an increase in *n*, or with a decrease in the coupling gap. Because the increased *n* reduces the difference of the refractive indices between the microdisk and background medium, more of the optical field will leak into the coupling gap. This results in an improvement in the overlap between the optical fields in two microdisks, and accordingly increases the value of *K*. Similarly, a smaller coupling gap also improves the overlap of the optical fields and provides greater coupling strength.

## 3. Optofluidic Microdisk Lasers

### 3.1. Fabrication of the Optofluidic Microdisk Device and Experimental Setup

The experimental setup of the optofluidic microcavity laser is shown in Figure 2a. The optofluidic microdisk device consists of a microfluidic channel and solid-state dye-doped polymer microdisks as shown in Figure 2b,c. Rhodamine B dye (RhB, CAS: 81-88-9, J&K Scientific Ltd., Beijing, China)-doped SU-8 photoresist (SU-8 2002, MicroChem Corp. MA, USA) in a ratio of 1.5 mg:1 mL was used as gain media and mixed by a magnetic stirrer for 2 h. During stirring, attention should be paid to shading the photoresist, in order to avoid quality deterioration. A 2.4 μm active layer of the gain media was then deposited by a spin coater (Specialty Coating Systems Inc., Indianapolis, USA) onto a silicon wafer with a 2 μm thick thermal oxide layer. The geometric shapes of the single and coupled microdisks were pre-designed on the chromium mask, and the gap between each pair of coupled microdisks was increased from 0.2 to 1 μm, and 0.1 μm for each pair. Therefore, the solid-state dye-doped polymer microdisk arrays were fabricated via single-mask standard lithography (Karl Suss MJB 3 Mask Aligner, SUSS MicroTec SE, Garching, Germany). The geometric sizes of the microdisks were measured via scanning electron microscopy (SEM), as shown in Figure 2c and Appendix A (details can be found in the Appendix A).

The optofluidic microdisk device consists of a microfluidic channel device and solid-state dye-doped polymer microdisks (details can be found in Appendix A). This microfluidic channel is a device that combines a clean glass slide and patterned polydimethylsiloxane (PDMS) leaf. The fabrication process of the optofluidic microdisk device is described in Appendix A. The shape of the glass mold was spliced with glass sheets, and the convex structure is shown as in Appendix A. After the PDMS leaf was formed from the glass mold, the inlet and outlet holes were punched by a needle. The solid-state dye-doped polymer microdisks were sandwiched between the microfluidic channel (PDMS leaf) and a glass slide, as shown in Appendix A. Finally, an optofluidic microdisk device, consisting of a microfluidic channel and solid-state dye-doped polymer microdisks, was fabricated, as shown in Figure 2b and Appendix A.

The laser spectra were measured using the experimental setup as shown in Figure 2a. The pump laser was a picosecond pulse laser (532 nm, PL2143A, EKSPLA, Vilnius, Lithuanian), and the pulse duration of the pump laser was 30 ps, with a repetition rate of 10 Hz. A monochromator (Acton SpectraPro 2750, Princeton Instruments, NJ, USA) was employed to collect the laser spectra. The spectra were measured by a monochromator with a 1200 grooves/mm grating and the spectral resolution was 0.01 nm. The size of the pump laser was controlled via tunable beam-shaping optics (BSO), and the pump laser energy was adjusted by a variable attenuator (VA). Firstly, the laser beam spot was coupled into the microscope system and further focused to nearly 50 μm under a 20× objective lens. Through the microscope with a camera, the pump position of the laser beam spot was able to be marked on a screen. The microcavity sample was then moved to the pump position using a 3-D stage. The output laser passed through a Glan–Taylor prism (GTP) and was collected by an optical multi-channel fiber bundle (core diameter 400 μm). The GTP was used as the polarization controller. The laser light was then coupled into the input slit (30 μm) of a monochromator (f = 0.75 m) and detected by an electron-multiplying charged coupled device (EMCCD, DV401A, Andor iDus, OX, UK). Finally, the laser spectra were recorded by a computer.

### 3.2. Modulation of the Microdisk Laser in Aqueous Media

A single microdisk was firstly investigated and the appearance of the microdisk sample was measured via SEM, as shown in Figure 3a. In this case, the diameter of the microdisk was set to 20.34 μm and the thickness was 2.4 μm, as measured using scanning electron microscopy (SEM) and a surface profiler, respectively. Compared to air cladding, the microdisk surrounded by water represents a much clearer spectrum. The water environment decreases the refractive index (RI) difference between the microdisk and the background media, where high-order WGMs with larger reflection angles at the cavity surface suffer from degradation of *Q* (quality) values and larger radiation losses. Therefore, they have a higher laser threshold and only fundamental-order WGMs with high *Q* values will exist in the laser spectra, which are shown in Figure 3b. The free spectral range (FSR) of this microdisk was 3.81 nm, agreeing well with the FSR calculated from *FSR* = *λ*^2^/(2*πRn_eff_*) ≈ 3.87 nm, where *λ* = 615 nm, *n_eff_* = 1.53, and *R* = 10.17 μm were the lasing wavelength, effective refractive index of the microdisk, and radius of the microdisk, respectively. The laser spectra were relatively clear in a water environment, which was conducive to sensing, single-frequency lasers, and other applications. Figure 3c,d show the simulation results of the electric field distribution of a single microdisk using the finite element method (FEM). To reduce consumption of computer memory, we divided the full 3-D model into two 2-D models, as shown in the top view and side view in Figure 3c,d, respectively. The refractive index of the microdisk in the simulation shown in Figure 3c was obtained by calculating the effective index of the three-layer planar waveguide of water-SU8-SiO_2_. As shown in Figure 3d, the electric field distribution in the cross section of the microdisk was calculated using an axially symmetric model [35]. All the simulation regions for the two 2-D models were surrounded by the perfect matching layer (PML) to avoid reflecting from the boundary. The microdisk was immersed in water, and the polarization of the fundamental-order-radial modes was transverse electric (TE) polarization.

The long-term stability of the microdisk laser was determined by pumping the microdisk every 20 s over a period of 1800 s, with the laser spectra being collected simultaneously. As shown in Figure 4a,b, the laser wavelength maintained the same position during this period. The standard deviation of the curve in Figure 4b is 5.84 pm, which is much smaller than the resolution of the monochromator (0.01 nm). These results indicate that an optofluidic microdisk laser does not have a large wavelength shift over a long term and that the system stability is good enough for our next experiments. 

The spectral modulation of the optofluidic microdisk laser can be achieved by changing the refractive index of the external environment, similar to the method for measuring the bulk refractive index sensitivity (*S*) of a sensor. Because this is an optofluidic device, it is essential to first characterize the sensitivity according to the changes in the refractive index of the solvent on the sensing surface [36]. In Figure 4c,d, the optofluidic microdisk laser was tested by flowing progressively higher concentrations of dimethyl sulfoxide (DMSO, CAS: 67-68-5, Lingfeng Chemical Reagent Co. Ltd., Shanghai, China) solutions over the sensing surface. The DMSO was diluted with deionized water (DI water) into concentrations of 2%, 4%, 6%, 8%, and 10%. The refractive indices of the DMSO solutions were proportional to the volume ratios of the DMSO (details can be found in Appendix A) [37]. Prior to use, the DI water and DMSO–water mixtures were kept at room temperature for a longer period, to minimize temperature-induced spectrum changes. A redshift of the wavelength in response to an increasing refractive index of the solution was observed from the laser spectra measurement and can be explained by *mλ* = 2*πn_eff_ R* (see details in Appendix A) [38]. *S_exp_* and *S_sim_* represent the *S* from the experiment and from the FEM simulation, respectively. *S_exp_* was obtained by directly measuring the wavelength shift, which was 18.14 nm/RIU. The process of detecting the *S* of the microdisk was also simulated using FEM, based on a 2-D axisymmetric model. Results show that the *S_sim_* of the fundamental and second-order radial modes with TE polarization were 18.12 and 28.90 nm/RIU, respectively, and that the *S_sim_* of the fundamental and second-order radial modes with transverse magnetic (TM) polarization were 22.98 and 37.65 nm/RIU, respectively. When the GTP was used to observe the experiment, the polarization state of the output laser was determined to be TE polarization. Therefore, the *S* of the measured microdisk was determined to be fundamental-order-radial mode with TE polarization.

### 3.3. Mode Splitting from the Coupled-Microdisk Laser in Aqueous Media

The method of purifying and adjusting the microcavity laser spectrum using the coupled microcavity structure is compact and effective. Normally, a typical WGM propagates along a spherical equatorial plane, whose evanescent field component allows interaction with the surrounding environment. Therefore, when two WGM microcavities are close to each other, their WGMs can be evanescently coupled and form mode splitting [31]. A large array of dye-doped polymer microlasers can be fabricated in parallel via a single step of deep-ultraviolet (DUV) lithography [39]. The coupled microdisks were investigated by immersing in water. At this time, the two modes experience the same gain or loss (*g_a_* = *g_b_*), and the frequency difference equals 2*κ*. Using the gap between the coupled microdisk of 0.3 μm, the diameters of these two coupled microdisks were 19.71 and 19.63 μm, and their SEM image is plotted in Figure 5a. Under evenly pumping schemes, as shown in the insert of Figure 5b, the mode splitting of the laser spectrum can be observed. Δ*λ* represents the distance between two laser peaks under mode splitting and is 0.183 nm in this case. Figure 5c shows the field distributions of WGMs with the fundamental-order-radial mode in a symmetric coupled-microdisk system. The wavelengths of the supermodes are 620.619 and 620.437 nm, respectively. Δ*λ* of the FEM simulation is 0.182 nm, which is very close to the experimental result of 0.183 nm. Different gaps between coupled microdisks were investigated under the same pumping conditions. Because the optical field decays exponentially outside the microdisk, the coupling strengths *κ* between the two cavities will decrease. Δ*λ* will also decrease, which is shown in Figure 5d. The experimental data and FEM simulation results were well-fitted by the exponential function. As the gap exceeded 0.6 μm, the coupling strengths *K* of the supermodes became weak and the Δ*λ* value stayed at a level of no more than a few dozen nanometers. Because of the limited resolution of the monochromator, narrower mode splitting cannot be observed in experiments.

Furthermore, when we change the refractive index of the aqueous medium *n*, it will also affect the coupling strengths of the supermodes *K*. Here, we used coupled microdisks with a gap of 0.4 μm. As shown in Figure 6a, when the refractive index of the DMSO solution was changed, the resonance detuning of the mode splitting was affected. Figure 6b shows that Δ*λ* of the experimental data was increased, which agreed well with the results of the FEM simulation. The increased background refractive index *n* will extract a greater part of the optical field into the coupling gap, leading to enhanced coupling strength and a larger difference in resonance frequencies of the coupled modes. The coupling coefficient *K* is as a function of *n_eff_*, which is given in Equation (5), and the experimental phenomenon is consistent with the FEM results in Figure 1b. 

In order to increase the difference in the gains/losses between the two microcavities, we used selective pumping schemes in which we changed the position of the pump beam spot gradually from the left side of the coupled microdisks to the right side by 8 μm per step. Figure 6 demonstrates the laser spectra of coupled microdisks with a 0.3 μm gap, with the coupled microdisks immersed in water. We focused on the wavelengths of the spectra from 612 to 615 nm, which are marked in red in Figure 7a,c. The laser spectra in Figure 7a,c correspond to the pump positions plotted in Figure 7b,d. The laser emerged when one of the coupled microdisks was pumped, and the splitting mode arose when the coupled microdisks were evenly pumped. When the pump beam spot gradually moved to the other side of the coupled microdisks, the splitting mode disappeared, and the laser finally vanished. This result is due to the selective pumping process, which changed the gain/loss status between the two coupled microdisks. The modulation gain branches of various supermodes will lead to a switchable single-frequency laser or mode splitting [9]. When only one of the coupled microdisks is pumped, the eigenfrequencies difference will become purely imaginary because in this case $\Delta \tilde{g}$ is large enough that the value under the square root in Equation (4) is smaller than zero. This will lead to the emission of a single-mode laser when the other laser modes are also below the threshold. Under even pumping, the gain difference between the two microcavities is very small and Δ*ω* is real, so it shows doublet peaks in the laser spectrum, i.e., mode splitting.

### 3.4. Single-Frequency Lasing from the Coupled Microdisk in Aqueous Media

Because of their high beam quality and spectral purity, lasers with a single-frequency laser emission feature are indispensable to many scientific and industrial applications, such as laser spectroscopy, laser metrology, and biomolecular sensing [5,6,39]. Normally, WGM lasers are usually multimode because of the lack of mode selection [9,40]. Methods such as microcavity size reduction, Vernier effect, and PT symmetry effect of coupled microcavities are possible strategies to realize single-frequency laser [9,14,40,41,42]. These methods need to control the gap size or thermal modulation of the two microcavities in order to control the coupling strength between them, which often requires a complex implementation system and results in low accuracy. 

The results of the previous section indicate that selective pumping schemes would change the spectrum of the coupled microdisks. In order to overcome the major drawback of a single-microcavity resonator laser, which essentially involves multimode laser emission, we used selective pumping schemes in a coupled-microdisk system, in order to realize a single-frequency laser emission. The coupled-microdisk resonator was dipped in water. In this case, the gap of the coupled microdisks was 0.5 μm. The position of the pump laser was “left pumping”, and the simulated mode field of the single-frequency laser is plotted in the inserts of Figure 8b. In the range 580–610 nm (much larger than one FSR and covers the tuning range of RhB in the ethanol solvent), only one laser peak has been collected by the spectrometer. Therefore, it is determined to be a single-frequency laser. The single-frequency laser spectra of the coupled microdisks at different pumping intensities are plotted in Figure 8a. The central wavelength of the single-frequency laser was 594.39 nm. When the pump laser energy was below the threshold, the spectrum was actually the fluorescence, which increases slowly with the pump energy. Moreover, the fluorescence efficiency of the gain media was very low. When the pump laser energy was higher than the laser threshold, the laser intensity will increase with the pump energy with a higher slope efficiency. On the curve in Figure 8b, there is a sudden change point (laser threshold) between the fluorescence and laser as the pump energy increases. The corresponding laser threshold curve shows that the laser threshold was 83.88 μJ/mm^2^, as shown in Figure 8b. Selective breaking of PT symmetry can systematically improve the effective amplification of single-mode operation [23]. 

An interesting phenomenon was observed when we changed the surrounding refractive index of the single-frequency laser. Another coupled-microdisk resonator was dipped in water under left pumping, with the gap at 0.6 μm. As shown in Figure 9a, the wavelength of the original single-frequency laser was 606.994 nm. With the increase in the liquid refractive index, the wavelength of the single-frequency laser gradually shifted to longer wavelengths, and the intensity decreased. Another hopped laser mode occurred at 603.131 nm. If the liquid refractive index is further increased, the original laser will disappear and only one laser peak can be seen in the spectrum. Figure 9c shows that *ln*(*I_hopped_/I_original_*) changed linearly with the liquid refractive index. Here, *I_original_* and *I_hopped_* are the light intensity of the original laser line and the hopped laser line, respectively. The hopped laser mode resulted from the slightly asymmetric structure of the coupled microdisks, which responded slightly differently to the change in the surrounding refractive index [30]. Therefore, the wavelength shifts of the coupled-microdisk resonances did not shift synchronously. Furthermore, the slight change of refractive index will also change the coupling efficiency of two coupled microdisks. As shown in Figure 9b,c, the wavelength shift sensitivities (*S_original_* and *S_hopped_*) were 14.04 and 14.65 nm/RIU, respectively, and the sensitivity of the intensity change (*S_I_*) was 149.87 RIU^−1^. Considering standard deviation (*σ* = 10 a.u.) of the background signal intensity as the lowest detectable lasing intensity, the detection limit of *S_I_* was 5.34 × 10^−5^ RIU, based on the equation in a previous article [30]. The resolution of the monochromator can reach 0.01 nm. The detection limits of *S_original_* and *S_hopped_* were 7.12 × 10^−^^4^ and 6.83 × 10^−4^ RIU, respectively. These results indicate that *S_I_* was 13 times more sensitive compared with *S_original_* and *S_hopped_*. Because of the Vernier effect, the coupling of the WGMs in the two resonators can produce a large amplification sensitivity. It was noted that the attenuation of light intensity was due to the bleaching of dye, caused by long-time laser pumping. The relative intensities of the original and hopped laser peaks were not affected. 

## 4. Conclusions

In this study, the spectral characteristics of coupled microdisks in aqueous media were examined. The spectral characteristics and modulation mechanism of mode splitting and single-frequency lasers from coupled microdisks were investigated. The resonance detuning of mode splitting was studied by changing the gap between two microdisks, the refractive index of the aqueous media, and the position of the pump laser, namely, selective pumping schemes. The variation of the single-frequency laser was studied by changing the intensity of the pump light and the refractive index of the aqueous media. For single-frequency lasers, the sensitivity can be amplified by changing the light intensity for sensing. The results of this study will help to deepen the understanding of the spectral characteristics and the modulation mechanism of microcavity lasers. Such optofluidic microcavity lasers have broad application prospects in tunable single-mode on-chip lasers and biosensors.

## Figures and Tables

**Figure 1 nanomaterials-09-01439-f001:**
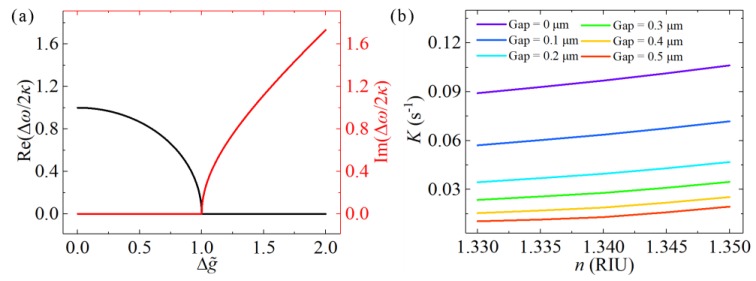
(**a**) Real and imaginary parts of Δ*ω*/2*κ* as a function of $\Delta \tilde{g}.$ The difference between the eigenfrequencies changes from purely real to purely imaginary at $\Delta \tilde{g}=0.$ (**b**) Coupling coefficient *K* as a function of refractive index under different gaps.

**Figure 2 nanomaterials-09-01439-f002:**
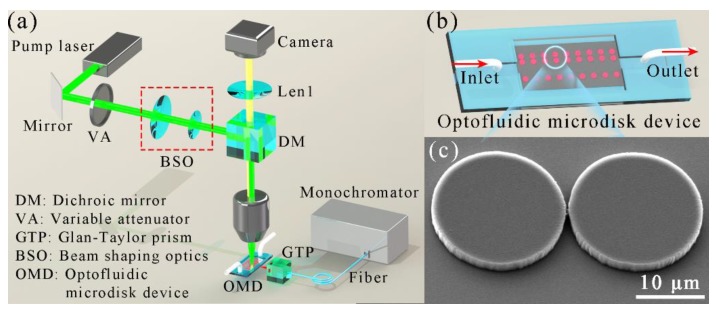
(**a**) Experimental setup schematic of the optofluidic microcavity laser. (**b**) 3-D schematic of the optofluidic microdisk device. (**c**) Scanning electron microscope (SEM) image of typical coupled microdisks. The scale bar is 10 μm.

**Figure 3 nanomaterials-09-01439-f003:**
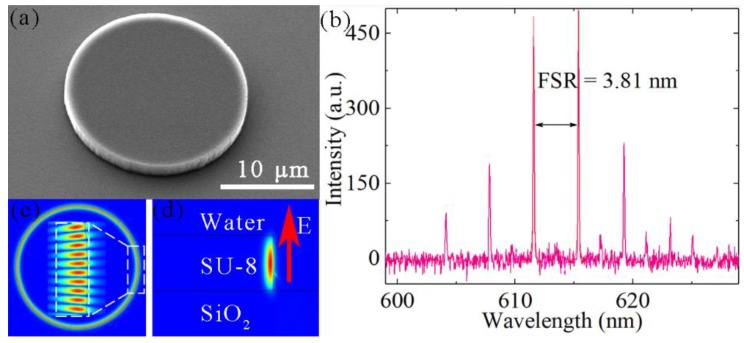
(**a**) SEM image of the microdisk. The scale bar is 10 μm. (**b**) Laser spectra of a microdisk dipped in water. The free spectral range (FSR) was 3.81 nm. (**c**) and (**d**) Field distributions of whispering-gallery modes (WGMs) in fundamental-order-radial mode with transverse electric (TE) polarization in the top view and side view. The direction indicated by the red arrow was the direction of electric field propagation. Finite element method (FEM) simulations were performed with the same parameters for the experimental data.

**Figure 4 nanomaterials-09-01439-f004:**
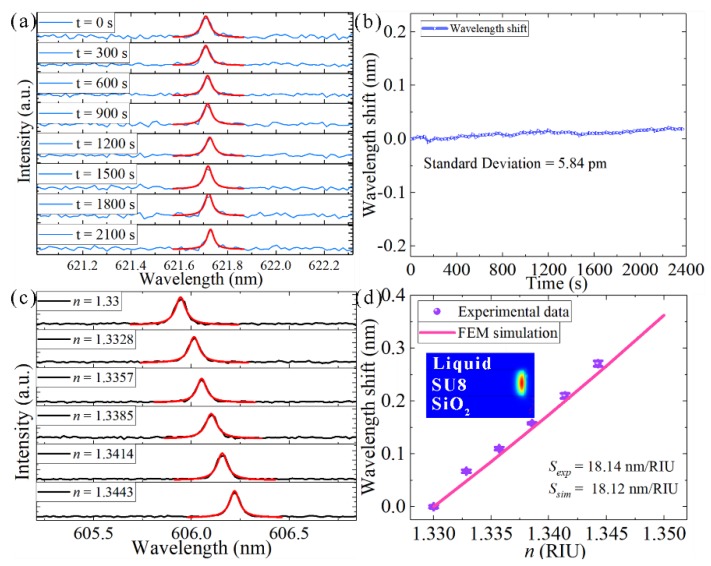
(**a**) Laser spectra of optofluidic microdisk laser at different times. (**b**) Measured wavelengths of the lasers as a function of time. Standard deviation was 5.84 pm. (**c**) Laser spectra shifts when the refractive index of the DMSO solution is slightly increased. (**d**) Mean wavelength shifts of the lasers as a function of the refractive indices of DMSO solutions. Violet dots and pink line represent the experimental data and FEM simulation data, respectively. Insert: fundamental-order-radial mode with TE polarization.

**Figure 5 nanomaterials-09-01439-f005:**
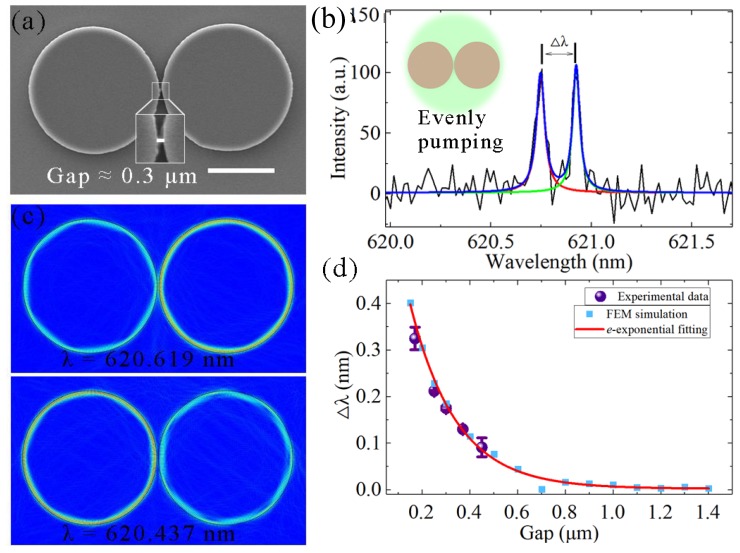
(**a**) SEM image of the coupled microdisks. Gap between two microdisks is approximately 0.3 μm. Scale bar is 10 μm. (**b**) Laser spectra of coupled microdisks with 0.3 μm gap. Δ*λ* denotes the distance between two peaks of mode splitting. Insert: schematic of evenly pumping schemes. (**c**) Field distributions of WGMs with fundamental-order-radial mode. (**d**) Δ*λ* of the experimental results and FEM simulations as a function of the gap. Purple dots represent the experimentally detected data; cerulean dots and red curves refer to the FEM simulation results and exponential fitting, respectively. FEM simulations were performed with the same parameters for the experimental data.

**Figure 6 nanomaterials-09-01439-f006:**
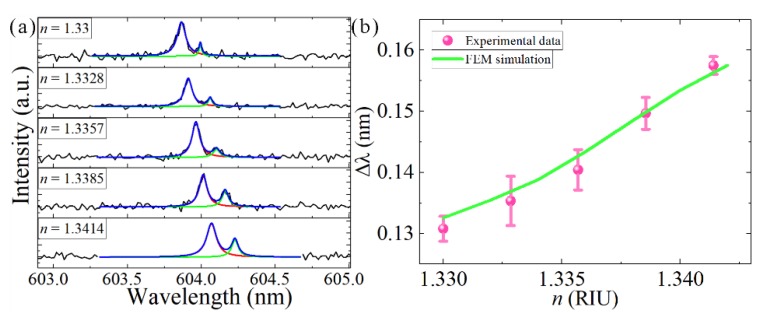
(**a**) Mode splitting in a coupled-microdisk resonator laser with resonance detuning. (**b**) Δ*λ* of the experimental data and FEM simulation, shown as a function of refractive index. Pink dots represent the experimentally detected data, while the green curves represent the FEM simulation results. FEM simulations were performed using the same parameters for the experimental data.

**Figure 7 nanomaterials-09-01439-f007:**
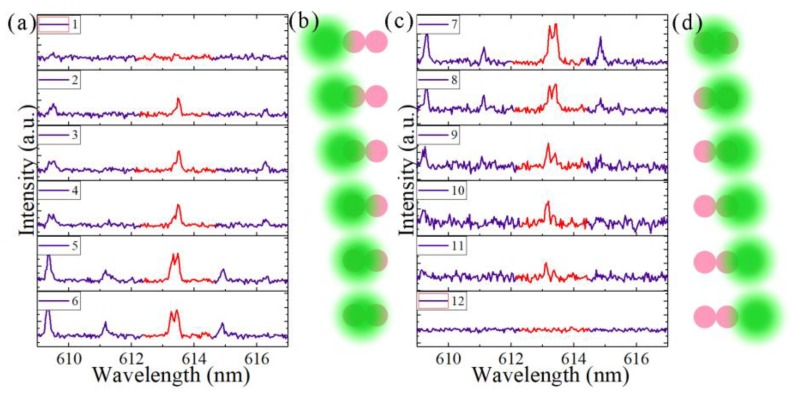
Laser spectra from different positions of the pump beam spot. (**a**) and (**c**) spectra during the moving of the pump laser from left to right. (**b**) and (**d**) schematics of the position of the pump beam spot.

**Figure 8 nanomaterials-09-01439-f008:**
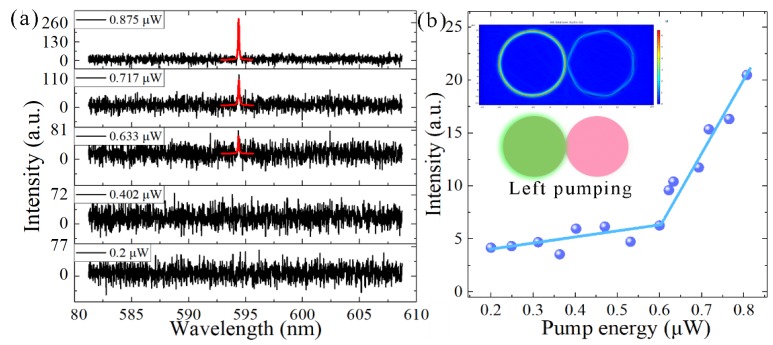
(**a**) Single-frequency emissions of the coupled microdisks under different pump energies. Insert: the energy of the pump laser. (**b**) Lasing intensity as a function of the pump energy intensity extracted from (**a**). Lasing threshold is approximately 0.60 μW (84.88 μJ/mm^2^) in a water environment. Insert: the position of the pump laser and the mode field of the single-frequency laser.

**Figure 9 nanomaterials-09-01439-f009:**
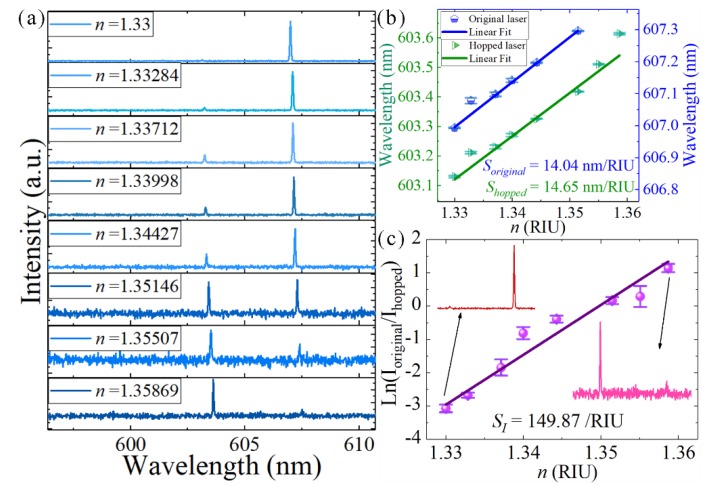
(**a**) Single-frequency emissions of the coupled-microdisk resonator laser. (**b**) Wavelength shifts of single-frequency lasers as a function of refractive index. (**c**) Intensity ratio [*ln*(*I_hopped_/I_original_*)] of the two lasing modes, as a function of refractive index. Insert: spectra at the two refractive indices values are given.

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
