# Peer review of "Spectral Modulation of Optofluidic Coupled-Microdisk Lasers in Aqueous Media"

_nanomaterials, 2019, doi:10.3390/nano9101439_

Round 1

Reviewer 1 Report

This manuscript clearly describes an optofluidic microcavity laser in aqueous solution. Also, the results indicate that the sensitivity of the intensity change is amplified owing to the Vernier effect. However, one question occurred to me in reading the article.

(1) In section 3.4, for single-frequency lasing from coupled microdisk in aqueous media, the selective breaking of PT symmetry was applied to your structure. However, as depicted in Figure 3., microdisk laser dipped in water has 3.81 nm free spectral range (FSR). Hence, to verify the single-mode operation, this fact needs to cover wide wavelength to check the side-mode, or explain how you confirm the results was in the single-frequency lasing.

Reviewer 2 Report

This work, concerning the modulation of optofluidic coupled microdisk lasers in aqueous media is real interesting and curious. Although authors made a significant set of experiences, some lack in deep explanations of the use of theoretical / physical models is one drawback. In addition, explanation of some results are not fully performed, particularly when we expect a more detailed connection between the theory in section 2.2 and all the final results. Finally, the grammar and sentences construction are not completely well done, leading to several misunderstanding of the specific ideas. Nevertheless the concept is interesting, some major corrections are needed before this manuscript can be accepted for publication in Nanomaterials. Specifically, the authors should consider the following points:

a) A deep English grammar and sentences needs to be made. Some parts of the manuscript are very difficult to understand (for instance the paragraph in lines 133-145 are absolutely understandable); also, some king of sentences like in line 138, “The shape of the microfluidic channel was like the Chinese characters ..”. This kind of sentences is not acceptable. Please, make a scheme is necessary.

b) In line 149, the authors says that “The grating of monochromator was adjusted to 1200g/mm”. First, the grating is not adjustable. The grooves are fixed by nature (or are the authors thinking in a different thing?); secondly, the grating specifications are grooves/mm and not g (gram?)/mm. There are others of this kind of mistakes across the manuscript;

c) Please specify clearly the conditions / assumptions and procedures of the FEM simulations

d) A more precise description about the experimental n(RIU) vs wavelength shift and n(RIU) vs K needs to be done because some experimental results seems to depart from the very known optical model.

d) In line 299 the authors says that “The corresponding laser threshold curve shows that the laser threshold was 83.88 uJ/mm2, as shown in Fig. 8(b)”. I understand that but the authors needs to explain clearly this conclusion, obtained from Fig 8b.

e) In line 267, the authors says that “The laser emerged when one of the coupled microdisks was pumped…This result is due to the selective pumping process, which changed the gain/loss status between the two coupled microdisks”. First, all these sentences needs to be clarified; and second, the last conclusion needs to be clear explained by any physical model.

f) How the authors can guarantee the precision of the refractive index? Any measurement to confirm the indicated data (or all of them are estimated?)

g) Finally, how the data presented by the authors can be compared with the literature? How is the improvement of this work?

Round 2

Reviewer 2 Report

This revised version of the manuscript seems to be clearly improved and more assertive in the “conduction line” of the work. Several explanation, regarding the concepts and particularly approximations made in the work, helps in an overall understand of the idea. I have checked carefully all the corrections and (again) the references. In general, the work still have a clear assessment of prime novelty but in general, as I above-mentioned, the quality has surely better. Moreover, some small English mistakes persists but the authors can re-check again. In this version, I agree that the manuscript can be published in Nanomaterials.

Author Response

We thank the Reviewer 2 for her/his careful reading of our manuscript again. This manuscript has been edited and polished again by MDPI English Editing (English editing ID: English-13018). We have made revisions of the manuscript carefully and checked for grammatical errors in manuscripts again. All changes made to the text are using the "Track Changes" function in the revised manuscript and Supplementary Materials. Please refer to the attached file.
